Preventative and therapeutic effects of a GABA transporter 1 inhibitor administered systemically in a mouse model of paclitaxel-induced neuropathic pain

Masocha Willias masocha@hsc.edu.kw
Parvathy Subramanian S.
Department of Pharmacology and Therapeutics, Faculty of Pharmacy, Kuwait University , Safat , Kuwait
Abdullah Jafri
Electronic publication date: 2016 Dec 15
Publication date: 2016
Volume: 4
Electronic Location ID: e2798
Received 2016 May 20; Accepted 2016 Nov 17
Copyright: ©2016 Masocha and Parvathy
Copyright year: 2016
Copyright holder: Masocha and Parvathy
License: This is an open access article distributed under the terms of the Creative Commons Attribution License, which permits unrestricted use, distribution, reproduction and adaptation in any medium and for any purpose provided that it is properly attributed. For attribution, the original author(s), title, publication source (PeerJ) and either DOI or URL of the article must be cited.
License URL: https://creativecommons.org/licenses/by/4.0/

Keywords: Chemotherapy-induced neuropathic pain, Paclitaxel, GABA transporter (GAT), GAT-1 inhibitor, Preventative treatment, Therapeutic treatment, Hyperalgesia, Motor coordination, Allodynia

Funding: Kuwait University Research Sector PT01/15 This study was supported by grant PT01/15 from Kuwait University Research Sector. The funders had no role in study design, data collection and analysis, decision to publish, or preparation of the manuscript.

==============================
Background

There is a dearth of drugs to manage a dose-limiting painful peripheral neuropathy induced by paclitaxel in some patients during the treatment of cancer. Gamma-aminobutyric acid transporter-1 (GAT-1) whose expression is increased in the brain and spinal cord during paclitaxel-induced neuropathic pain (PINP) might be a potential therapeutic target for managing PINP. Thus, our aim was to evaluate if systemic administration of a GAT-1 inhibitor ameliorates PINP.

Methods

The reaction latency to thermal stimuli (hot plate test; at 55 °C) and cold stimuli (cold plate test; at 4 °C) of female BALB/c mice was recorded before and after intraperitoneal treatment with paclitaxel, its vehicle, and/or a selective GAT-1 inhibitor NO-711. The effects of NO-711 on motor coordination were evaluated using the rotarod test at a constant speed of 4 rpm or accelerating mode from 4 rpm to 40 rpm over 5 min.

Results

The coadministration of paclitaxel with NO-711 3 mg/kg prevented the development of paclitaxel-induced thermal hyperalgesia and cold allodynia at day 7 after drug treatment. NO-711 at 3 mg/kg produced antihyperalgesic activity up to 1 h and antiallodynic activity up to 2 h in mice with established paclitaxel-induced thermal hyperalgesia and cold allodynia. No motor deficits were observed with NO-711 at a dose of 3 mg/kg, whereas a higher dose 5 mg/kg caused motor impairment and reduced mean time spent on the rotarod at a constant speed of 4 rpm. However, at a rotarod accelerating mode from 4 rpm to 40 rpm over 5 min, NO-711 3 mg/kg caused motor impairment up to 1 h, but had recovered by 2 h.

Conclusions

These results show that systemic administration of the GAT-1 inhibitor NO-711 has preventative and therapeutic activity against paclitaxel-induced thermal hyperalgesia and cold allodynia. NO-711’s antiallodynic effects, but not antihyperalgesic effects, were independent of its motor impairment/sedation properties. Thus, low doses of GAT-1 inhibitors could be useful for the prevention and treatment of PINP with proper dose titration to reduce motor impairment/sedation side effects.

Introduction

Chemotherapy-induced neuropathic pain (CINP) limits the use of some chemotherapeutic drugs, such as paclitaxel, oxaliplatin and vincristine, in the management of various types of cancer. The incidence of chemotherapy-induced peripheral neuropathy in patients treated with some chemotherapeutic drugs is very high, for example in patients treated with paclitaxel it is around 70.8% (95% CI [43.5–98.1]) (Seretny et al., 2014). In another study it was found that 64% of patients experienced chemotherapy-induced peripheral neuropathy during paclitaxel treatment and 27% of these patients were diagnosed with neuropathic pain (Reyes-Gibby et al., 2009). Unfortunately, there is a dearth of drugs to prevent or manage this type of pain. Currently, only duloxetine has a moderate recommendation for the management of CINP, whilst other drugs used for other neuropathic pain conditions may be given because of the limited CINP treatment options (Hershman et al., 2014). Thus, studies on the pathophysiology of CINP and the development of new treatment options are essential.

Using an animal model of paclitaxel-induced neuropathic pain (PINP) we recently observed an increased expression of gamma-aminobutyric acid transporter 1 (GAT-1) transcripts in the anterior cingulate cortex (ACC) (Masocha, 2015); which is an area involved in pain perception and modulation (Seminowicz et al., 2009; Steenland et al., 2006; Xie, Huo & Tang, 2009). GAT-1 is responsible for most of the GABA uptake from the synaptic cleft in the brain (Borden, 1996; Conti et al., 1998; Jensen et al., 2003). In the same year (2015) another group also reported increased expression of GAT-1 in the spinal cord of an animal model of PINP (Yadav et al., 2015). In addition, they observed that intrathecal injection of a GAT-1 inhibitor ameliorates PINP (Yadav et al., 2015). Thus, suggesting that GAT-1 plays an important role in the pathophysiology of PINP and is a potential therapeutic target.

Tiagabine, a GAT-1 inhibitor, is used for the treatment of epilepsy as an add-on therapy in the treatment of partial seizures (Bialer et al., 2007). It has various adverse effects of which the most common include fatigue, dizziness, psychomotor slowing, ataxia, somnolence, insomnia, nausea/GI upset and weight change (Bialer et al., 2007; Vossler et al., 2013). Tiagabine has also been shown to have some beneficial effects for the treatment of neuropathic pain (Novak et al., 2001). In animal models of neuropathic pain NO-711, a GAT-1 inhibitor, has been reported to have antiallodynic and antihyperalgesic activity (Daemen et al., 2008; Yadav et al., 2015). NO-711 is a selective and the most potent non-competitive inhibitor of GAT-1 that can cross the blood–brain barrier (Borden et al., 1994; Hog et al., 2006; Soudijn & Van Wijngaarden, 2000). However, in one study it was suggested that the observed antinociceptive activity of the GAT-1 inhibitor tiagabine could be attributed to sedative and motor-impairing properties, as these properties can produce false positive effects in some pain tests (Salat et al., 2015).

The aim of this study was to evaluate whether systemic administration of a GAT-1 inhibitor can prevent the development of PINP and also if it has therapeutic effects against established PINP at doses that do not impair motor activity.

Materials and Methods

Animals

Animals used in this study were handled in compliance with the Kuwait University, Health Sciences Center (HSC), Animal Resources Centre (ARC) guidelines and in compliance with Directive 2010/63/EU of the European Parliament and of the Council on the protection of animals used for scientific purposes. All animal experiments were approved by the Ethical Committee for the use of Laboratory Animals in Teaching and in Research, HSC, Kuwait University. Female BALB/c mice (8 to 12 weeks old; 20–30 g; n = 247) supplied by the ARC at the HSC, Kuwait University were used in this study. The animals were kept in temperature controlled (24 ± 1 °C) rooms with food and water given ad libitum.

Administration of paclitaxel to induce thermal hyperalgesia and cold allodynia

Paclitaxel (Tocris, Bristol, UK) was dissolved in a solution made up of 50% Cremophor EL and 50% absolute ethanol to a concentration of 6 mg/ml and then diluted in normal saline (NaCl 0.9%), to a final concentration of 0.2 mg/ml just before administration. Vehicle or paclitaxel 2 mg/kg were injected intraperitoneally (i.p.) for five consecutive days, the cumulative dose of paclitaxel was 10 mg/kg (the paclitaxel administration schedule is depicted in Fig. 1). This paclitaxel treatment regimen produces painful neuropathy and thermal hyperalgesia in mice (Nieto et al., 2008; Parvathy & Masocha, 2013).

Figure 1 Drug administration schedule for preventative and therapeutic treatment with NO-711 against paclitaxel-induced thermal hyperalgesia and cold allodynia.

(A) Treatment with NO-711 to prevent the development of paclitaxel-induced thermal hyperalgesia and cold allodynia. (B) Therapeutic treatment with NO-711 to alleviate established paclitaxel-induced thermal hyperalgesia and cold allodynia. The arrows indicate the days when the drugs were intraperitoneally administered.

Administration of NO-711

1-[2-[[(diphenylmethylene)imino]oxy]ethyl]-1,2,5,6-tetrahydro-3-pyridinecarboxylic acid hydrochloride (NO-711) (Sigma, St. Louis, MO, USA) was dissolved in normal saline and administered to mice i.p. at a volume of 10 ml/kg body mass.

Two treatment regimens were used to treat paclitaxel-treated mice; the first was a preventative/prophylactic treatment (Fig. 1A) and the second a therapeutic treatment (Fig. 1B). The doses of NO-711 were chosen based on those previously shown to have antinociceptive activity in mice (Kubo et al., 2009).

For the preventative treatment NO-711 3 mg/kg was co-administered daily with paclitaxel, as described above, for five consecutive days (Fig. 1A). The mice were assessed for the development of thermal hyperalgesia and cold allodynia on day 7 and those that received paclitaxel plus NO-711 were compared with the mice treated with the paclitaxel plus vehicle (for NO-711) only.

For the therapeutic treatment, NO-711 at doses of 1 mg/kg and 3 mg/kg, was administered once at seven days after first administration of paclitaxel (Fig. 1B), when mice had developed thermal hyperalgesia, as previously described (Parvathy & Masocha, 2013), and cold allodynia.

For the rotarod test NO-711 was administered once at doses of 3 mg/kg and 5 mg/kg to naïve mice.

Assessment of thermal nociception

Reaction latencies to the hot-plate test were measured as described previously (Parvathy & Masocha, 2013), before (baseline latency), at day 7 after first injection of paclitaxel alone or together with NO-711 (preventative treatment), and at various times on day 7 starting at 30 min after treatment with NO-711 (therapeutic treatment). Briefly, mice were individually placed on a hot plate (Panlab SL, Barcelona, Spain) with the temperature adjusted to 55 ± 1 °C. The time to the first sign of nociception, paw licking, flinching or jump response to avoid the heat was recorded and the animal immediately removed from the hot plate. A cut-off period of 20 s was maintained to avoid damage to the paws. The observer was blinded to the treatment the animal received.

Assessment of cold allodynia

Reaction latencies to the cold-plate test were measured as described previously (Sanna et al., 2014), before (baseline latency), at day 7 after first injection of paclitaxel alone or together with NO-711 (preventative treatment), and at various times on day 7 starting at 30 min after treatment with NO-711 (therapeutic treatment). Briefly, mice were individually placed on a cold plate (Panlab SL, Barcelona, Spain) with the temperature adjusted to 4 ± 1 °C. The time it took for the animal to lick its paws was recorded and the animal immediately removed from the cold plate. A cut-off period of 60 s was maintained to avoid damage to the paws. The observer was blinded to the treatment the animal received.

Assessment of motor coordination

Motor coordination was evaluated using the rotarod apparatus (Panlab SL, Barcelona, Spain) using two protocols. In the first protocol the rotation of the rod was set at a constant speed of 4 rpm. In the second protocol an accelerating mode from 4 rpm to 40 rpm over 5 min was used, as described previously (Galante et al., 2009). All animals were trained for 3 days. On the test day, mice received single injections of NO-711 3 and 5 mg/kg or its vehicle (normal saline) before the test. The latency (in seconds) for the first fall was recorded at 30 min, 1 h and 2 h after administration of NO-711. The cut-off time was set at 300 s. The observer was blinded to the treatment the animal received.

Statistical analyses

Statistical analyses were performed using Student’s t-test, one-way analysis of variance (ANOVA) followed by Newman-Keuls post-tests or two-way repeated measures ANOVA followed by Bonferroni post-tests using GraphPad Prism software (version 5.0). The differences were considered significant at p < 0.05. The results in the text and figures are expressed as the means ± S.E.M.

Results

Paclitaxel-induced thermal hyperalgesia and cold allodynia

Mice treated with paclitaxel developed thermal hyperalgesia i.e., had significant lower reaction latency times (about 27% lower) to the hot plate (55 °C) on day 7 compared to vehicle-treated mice, 7.1 ± 0.3 s versus 9.7 ± 0.2, respectively (p < 0.05; Fig. 2A), similar to what we described previously (Parvathy & Masocha, 2013).

Figure 2 Paclitaxel-induced thermal hyperalgesia and cold allodynia in BALB/c mice.

(A) Thermal hyperalgesia in BALB/c mice at day 7 post first inoculation of paclitaxel in a hot plate test (55 °C). Each point represents the mean ± S.E.M of values obtained from 15 vehicle-treated and 15 paclitaxel-treated animals. (B) Cold allodynia in BALB/c mice at day 7 post first inoculation of paclitaxel in a cold plate test (4 °C). Each point represents the mean ± S.E.M of values obtained from 12 vehicle-treated and 13 paclitaxel-treated animals. **p < 0.01 compared to drug vehicle at the same day after treatment and ##p < 0.01 compared to pretreatment (PRE) values (Student’s t test).

Mice treated with paclitaxel also developed cold allodynia i.e., had significant lower reaction latency times (about 31% lower) to the cold plate (4 °C) on day 7 compared to vehicle-treated mice, 38.4 ± 3.2 s versus 55.7 ± 2.2, respectively (p < 0.05; Fig. 2B).

NO-711 prevents the development of paclitaxel-induced thermal hyperalgesia and cold allodynia

Mice treated with paclitaxel had significant lower reaction latency times to the hot plate (55 °C) on day 7 compared to vehicle-treated mice, 6.3 ± 0.3 s versus 9.5 ± 0.4, respectively (p < 0.05; Fig. 3A). On the other hand, mice treated with paclitaxel plus NO-711 (3 mg/kg) consecutively for 5 days had reaction latency times similar to the vehicle-only treated control mice, 8.7 ± 0.4 s versus 9.5 ± 0.4, respectively (p > 0.05), which were significantly higher than those of the mice treated with paclitaxel plus vehicle (p < 0.01; Fig. 3A).

Mice treated with paclitaxel had significant lower reaction latency times to the cold plate (4 °C) on day 7 compared to vehicle-treated mice, 32.3 ± 1.7 s versus 58.7  ± 0.4, respectively (p < 0.05; Fig. 3B). On the other hand, mice treated with paclitaxel plus NO-711 (3 mg/kg) consecutively for 5 days had reaction latency times similar to the vehicle-only treated control mice, 51.2 ± 3.3 s versus 59.7 ± 0.3, respectively (p > 0.05), which were significantly higher than those of the mice treated with paclitaxel plus vehicle (p < 0.01; Fig. 3B).

Figure 3 Coadministration of NO-711 with paclitaxel protects against the development of paclitaxel-induced thermal hyperalgesia and cold allodynia.

(A) Effects of coadministration of paclitaxel with NO-711 on the development of paclitaxel-induced thermal hyperalgesia in BALB/c mice in a hot plate test (55 °C). Each point represents the mean ± S.E.M of the values obtained from 8 animals. (B) Effects of coadministration of paclitaxel with NO-711 on the development of paclitaxel-induced cold allodynia in BALB/c mice in a cold plate test (4 °C). Each point represents the mean ± S.E.M of the values obtained from 16–29 animals **p < 0.01 compared to control mice (treated with vehicles only) and ##p < 0.01 compared to mice treated with paclitaxel + vehicle (two-way repeated measures ANOVA followed by Bonferroni post-test).

NO-711 alleviates established paclitaxel-induced thermal hyperalgesia and cold allodynia

Mice with paclitaxel-induced thermal hyperalgesia and cold allodynia (i.e., mice with significantly lower reaction times after treatment with paclitaxel compared to pretreatment values) were treated with two doses of NO-711, 1 and 3 mg/kg.

Figure 4 Antihyperalgesic and antiallodynic effects of NO-711 on BALB/c mice with paclitaxel-induced thermal hyperalgesia and cold allodynia.

(A) Reaction latency times (taken before (PRE) and at day 7 post first administration of paclitaxel) at different times after treatment with NO-711 (1 and 3 mg/kg) or its vehicle in a hot-plate test (55 °C). Each bar represents the mean  ± S.E.M of values obtained from 8–16 animals. (B) Reaction latency times (taken before (PRE) and at day 7 post first administration of paclitaxel) at different times after treatment with NO-711 (1 and 3 mg/kg) or its vehicle in a cold-plate test (4 °C). Each bar represents the mean ± S.E.M of values obtained from 5–7 animals. **p < 0.01 compared to drug vehicle at the same time point after treatment (two-way repeated measures ANOVA followed by Bonferroni test); #p < 0.5, ##p < 0.01 compared to day 7 post paclitaxel values before NO-711 administration; and $p < 0.5, $$p < 0.01 compared to values before administration of paclitaxel (Pre-paclitaxel) (one-way ANOVA followed by Newman-Keuls post-test).

The intraperitoneal administration of vehicle or a lower dose of NO-711 (1 mg/kg) did not change the reaction latency to thermal stimuli (55 °C) in mice with paclitaxel-induced thermal hyperalgesia compared to before vehicle or NO-711 1 mg/kg administration at day 7 (p > 0.05; Fig. 4A). However, NO-711 at a dose of 3 mg/kg produced significant increase in reaction latency in mice with paclitaxel-induced thermal hyperalgesia at 30 min and 1 h post drug administration compared to mice treated with vehicle and before NO-711 administration at day 7 (p < 0.01) but had not effect at 2 h post drug administration (p < 0.05; Fig. 4A). The mice with paclitaxel-induced thermal hyperalgesia treated with NO-711 at a dose of 3 mg/kg had reaction latency at 30 min and 1 h that was not significantly different to the reaction latency before paclitaxel administration to mice, 10.1 ± 0.6 s, 8.8 ± 0.6 s and 9.7 ± 0.3 s respectively (p >0.05) but was significantly lower at 2 h 7.8 ± 0.4 s (p < 0.05; Fig. 4A).

The intraperitoneal administration of vehicle or a lower dose of NO-711 (1 mg/kg) did not change the reaction latency to cold stimuli (4 °C) in mice with paclitaxel-induced cold allodynia compared to before vehicle or NO-711 1 mg/kg administration at day 7 (p > 0.05; Fig. 4B). However, NO-711 at a dose of 3 mg/kg produced significant increase in reaction latency in mice with paclitaxel-induced cold allodynia at all times measured 30 min, 1 h and 2 h, post drug administration compared to mice treated with vehicle and before NO-711 administration at day 7 (p < 0.01; Fig. 4B). The mice with paclitaxel-induced cold allodynia treated with NO-711 at a dose of 3 mg/kg had reaction latency at 30 min, 1 h and 2 h that was not significantly different to the reaction latency before paclitaxel administration to mice, 60 s, 60 s, 56.2 ± 3.6 s and 57.9 ± 0.8 s respectively (p > 0.05; Fig. 4B).

Motor coordination

Side effects of GAT-1 inhibitors such as tiagabine include sedation and impairment of motor coordination (Salat et al., 2015). Impairment of motor coordination and sedation affects the results of behavioural tests, including the hot plate and cold plate tests. Thus the effect of NO-711 on motor coordination was evaluated using the rotarod test. No significant differences of the mean time spent on the rotarod were observed between mice treated with vehicle (300 s) and the mice treated with NO-711 3 mg/kg (265 ± 35 s, 269 ± 30 s and 300 s at 30 min, 1 h and 2 h after drug administration, respectively (p > 0.05; Fig. 5A) at a rotarod constant speed of 4 rpm. However, mice treated with a higher dose of NO-711, 5 mg/kg, had a significant decrease in the mean time spent on the rotarod from 300 s to 123 ± 52 s and 185 ± 35 s at 30 min and 1 h after drug administration, respectively (p < 0.01; Fig. 5A), indicating motor impairment, but had recovered by 2 h, 300 s (p > 0.05). When NO-711 3 mg/kg was evaluated at a rotarod accelerating mode from 4 rpm to 40 rpm over 5 min, it had a significant decrease in the mean time spent on the rotarod from 68.8 ± 3.8 s to 23.8 ± 5.4 s and 48.6 ± 4.3 s at 30 min and 1 h after drug administration, respectively (p < 0.01; Fig. 5B), indicating motor impairment, but had recovered by 2 h, 65.9 ± 3.6 s (p > 0.05).

Figure 5 Time course of the mean time spent on the rotarod (s) for NO-711 in a rotarod test in naïve BALB/c mice.

(A) At a constant speed of 4 rpm. Each point represents the mean  ± S.E.M of values obtained from 8 animals. (B) At an accelerating mode from 4 rpm to 40 rpm over 5 min. Each point represents the mean ± S.E.M of values obtained from 12–13 animals. **p < 0.01 compared to drug vehicle at the same time point after treatment (two-way repeated measures ANOVA followed by Bonferroni post-test).

Discussion

The results of this study show that systemic administration of NO-711, a GAT-1 inhibitor, prevents the development of paclitaxel-induced thermal hyperalgesia and cold allodynia and also has antihyperalgesic and antiallodynic activity in mice with established paclitaxel-induced thermal hyperalgesia and cold allodynia. However, the antihyperalgesic effects of NO-711 could not be separated from its sedative and motor-impairing properties, whereas its antiallodynic effects were independent of its sedative and motor-impairing properties.

GAT-1 inhibitors have anti-seizure activities and one of them, tiagabine, is used for the treatment of epilepsy as an add-on therapy in the treatment of partial seizures (Bialer et al., 2007). They have also been shown to have antinociceptive and antiallodynic activities (Daemen et al., 2008; Ipponi et al., 1999; Li et al., 2011; Yadav et al., 2015). Subcutaneous administration of NO-711 produced antiallodynic activities in rats with CCI-induced neuropathic pain (Daemen et al., 2008). Intrathecal administration of NO-711 produced antiallodynic activities rats with CCI-induced neuropathic pain and in rats with PINP (Li et al., 2011; Yadav et al., 2015). On the other hand, intraperitoneal administration of another GAT-1 inhibitor SKF89976A did not have effect on established allodynia in a partial sciatic nerve ligation (PSL) mouse model (Jinzenji et al., 2014). Intraperitoneal administration of tiagabine reversed SCI-induced neuropathic pain (Meisner, Marsh & Marsh, 2010). However, in one study it was suggested that the observed antinociceptive activity of tiagabine could be attributed to sedative and motor-impairing properties, as these properties can produce false positive effects in some pain tests (Salat et al., 2015). In the rotarod test we observed that NO-711 at a dose of 3 mg/kg did not show significant motor impairment at low speed (4 rpm) of the rotarod but showed motor impairment/sedation up to 1 h at an accelerating mode (4–40 rpm) of the rotarod, which is in contrast to what has been described before for that dose i.e., it did not cause motor/hypnotic effects (Kubo et al., 2009). A higher dose, 5 mg/kg, caused motor impairment/sedation, even at low speed (4 rpm) on the rotarod, similar to what has been observed with other high doses of NO-711 (Kubo et al., 2009; Suzdak et al., 1992). Thus, NO-711 has a dose-dependent motor impairment/sedation effect as described previously (Suzdak et al., 1992). In order to separate antinociceptive, antihyperalgesic or antiallodynic effects of NO-711 from motor impairment/sedation effect a dose of 3 mg/kg was used for evaluation of the activity of the compound against paclitaxel-induced thermal hyperalgesia and cold allodynia, up to 2 h, a time point when the drug had no motor impairment/sedation effects.

Intrathecal administration of NO-711 has recently been shown to attenuate established paclitaxel induced mechanical and thermal hyperalgesia in rats without causing motor impairment or sedation (Yadav et al., 2015). In the current study systemic (intraperitoneal) administration of NO-711 at a dose of 3 mg/kg attenuated established paclitaxel-induced thermal hyperalgesia and cold allodynia in mice. However, the effect of NO-711 on paclitaxel-induced thermal hyperalgesia could not be separated from its motor impairment/sedation effects because they both lasted up to 1 h after NO-711 administration. On the other hand, the effect of NO-711 on paclitaxel-induced cold allodynia could be separated from its motor impairment/sedation effects because motor impairment lasted up to 1 h after NO-711 administration, whereas the drug had antiallodynic activity at 2 h when it did not have motor impairment/sedation effects. No other studies have reported the effects of a systemically administered GAT-1 inhibitor on PINP. However, systemic administration of GAT-1 inhibitors has been reported to attenuate thermal hyperalgesia and allodynia in other models of neuropathic pain such as chronic constriction injury of the sciatic nerve, spinal nerve ligation (Daemen et al., 2008; Giardina et al., 1998). Thus, systemic administration of NO-711 has antiallodynic activities against PINP similar to the systemic administration of another GAT-1 inhibitor, tiagabine, in other models of neuropathic pain.

Our findings and those Yadav and colleagues showing increased GAT-1 expression in the CNS, in the ACC and spinal cord, respectively, suggests an important role of GAT-1 in the pathophysiology of PINP (Masocha, 2015; Yadav et al., 2015). Thus, we explored whether preventative treatment by inhibiting GAT-1 activity could be potentially useful to prevent the development of PINP. Our findings show that coadministration of NO-711 at a dose of 3 mg/kg with paclitaxel for five consecutive days prevented the development of paclitaxel-induced thermal hyperalgesia and cold allodynia. Thus, indicating that increased GAT-1 expression and activity play a role in the development of PINP. The possible mechanism of GAT-1 inhibitors in PINP is possibly through restoration of ambient GABA levels. The increased GAT-1 expression during PINP consequently increases GABA uptake and thus reduces ambient GABA levels and decreases inhibitory tone. Recently, we reported that there is increased excitability in the ACC because of a deficiency in GABA-mediated neurotransmission in that brain area and addition of exogenous GABA reverses the increased excitability in the ACC induced by paclitaxel (Nashawi et al., 2016).

Conclusions

In conclusion our results show that systemic administration of a GAT-1 inhibitor, NO-711, prevents the development of paclitaxel-induced thermal hyperalgesia and cold allodynia and alleviates established paclitaxel-induced thermal hyperalgesia and cold allodynia. The antiallodynic activity of the GAT-1 inhibitor is independent of its motor impairment/sedative activities, whereas its antihyperalgesic activity are not. Thus, low doses of GAT-1 inhibitors have potential therapeutic activity to prevent or manage PINP and CINP in general with proper dose titration to reduce motor impairment/sedation side effects. Therefore the possible clinical use of GAT-1 inhibitors, which are already in the clinics such as tiagabine, against CINP warrants further research.

Supplemental Information

Supplemental Information 1 Thermal hyperalgesia and cold allodynia in BALB/c mice at day 7 post first inoculation of paclitaxel

Click here for additional data file.

Supplemental Information 2 Effects of coadministration of paclitaxel with NO-711 on the development of paclitaxel-induced thermal hyperalgesia and cold allodynia in BALB/c mice

Click here for additional data file.

Supplemental Information 3 Reaction latency times (s) (taken at day 7 post first administration of paclitaxel) at different times after treatment with NO-711 (1 and 3 mg/kg) or its vehicle in the hot-plate and cold-plate tests

Click here for additional data file.

Supplemental Information 4 Time course of the mean time spent on the rotarod (s) after NO-711 treatment in a rotarod test in naïve BALB/c mice

Click here for additional data file.

We are grateful to Ms. Amal Thomas for her technical assistance and the staff from the Animal Resources Centre, HSC, Kuwait University for their support.

Additional Information and Declarations

Competing Interests

Author Contributions

Animal Ethics

Data Availability

The authors declare there are no competing interests.

Willias Masocha conceived and designed the experiments, analyzed the data, contributed reagents/materials/analysis tools, wrote the paper, prepared figures and/or tables, reviewed drafts of the paper.

Subramanian S. Parvathy performed the experiments, analyzed the data, reviewed drafts of the paper.

The following information was supplied relating to ethical approvals (i.e., approving body and any reference numbers):

All animal experiments were approved by the Ethical Committee for the use of Laboratory Animals in Teaching and in Research, HSC, Kuwait University.

The following information was supplied regarding data availability:

The raw data has been supplied as a Supplemental Dataset.

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
