# Peer review of "Preventative and therapeutic effects of a GABA transporter 1 inhibitor administered systemically in a mouse model of paclitaxel-induced neuropathic pain"

_PeerJ, doi:10.7717/peerj.2798_

## Round 0.1 · original submission · Major Revisions

Dear Authors,

Please carefully and meticulously revised your manuscript according to the comments of ALL peer reviewers as this is very important to ensure that it will not be rejected in the next round of re- submission.

Reviewer 1 ·

Basic reporting

No Comments

Experimental design

No Comments

Validity of the findings

No Comments

Additional comments

In this study, the authors demonstrated that systemic administration of the GAT-1 inhibitor NO-711 has preventative and therapeutic activity against paclitaxel-induced thermal hyperalgesia. While the manuscript was well written and the conclusion was properly drawn, this study is lack of innovation and mechanistic investigation. A similar study with more mechanistic investigation was just recently published (Yadav et al 2015). In my review, data collected in this study is not efficient and innovative to be published.

Reviewer 2 ·

Basic reporting

Author used very professional English language throughout the manuscript. Intro and background to show context were sufficient to make the relevance case to investigate. Literature well referenced and related to the investigation. Figures are relevant, high quality, well labelled along clarity and statistical analysis.

Experimental design

Author used the well established method in this field to verify theirs findings. There was clear need to show the NO-711 (GAT-1 Inhibitor) effects on the motor function. methods described by authors are sufficient and enough to replicate. Although couple of study has mentioned about the NO-711 motor effects but in this study authors have provided the raw data in the main article for others to make observation.

Validity of the findings

Data is robust, statically sound and along with relevant control controlled groups.

Additional comments

I am curious if any western blot is done to see the GAT-1 transporter protein expression in a preventive group (NO-711 + Paclitaxel). This data could shed the light if there is negative feed back mechanism in the regulation of GAT-1 transporter expression in PINP. Western blot analysis could further provide the molecular evidence to support the chronic NO-711 treatment effects and demonstrate its direct effects due to the GAT-1 transporter.

·

Basic reporting

The authors do not specifically mention where they are blinded to the experimental conditions in the behavioral experiments.

Experimental design

No comments

Validity of the findings

The results are robust but there is one issue with respect to the clinical relevance. The primary complaints of PIPN or CIPN patients is mechanical allodynia, cold hypersensitivity and ongoing pain. The authors measured heat hyperalgesia. This is suboptimal for this model. It would be interesting to know if the GAT-1 inhibitor also worked on mechanical and cold hypersensitivity.

Additional comments

The authors have looked at the effect of a GAT-1 inhibitor on chemotherapy induced pain in a rodent model. They find an antinociceptive effect that is separable from motor impairment. Importantly, the clinically available drug that they used is able to reverse established CIPN. There are some problems with the experimental endpoints that are addressed above. I have two other points:
1) The authors do not make any comments in the discussion about the probably mechanism of action of GAT-1 inhibitors in PINP. It would be appropriate for the authors to mention this. Presumably the increased GAT-1 expression reduces ambient GABA levels and decreases inhibitory tone. Is this what the authors think could be happening?
2) The authors do not specify how this study differs from the work of Yadav et al. Is the difference simply that in the present study the drug was given systemically?
With these comments in mind, I close by stating that this is simple but nicely done study that could have a clinical impact.

Reviewer 4 ·

Basic reporting

In general, this is an interesting paper. The idea of testing new chemical entities (NO-711) and novel drug targets (GAT1) in neuropathic pain which is ofter drug-resistant, is very important. The English language sounds good, the structure of the manuscript requires some re-organisation..
Before being published, some corrections are necessary.
1. Introduction: a short description of pharmacological properties of NO-711 is missing.
2. Methods:
- What was the rationale for the selection of the doses 1, 3 and 5 mg/kg for the experiments?
- The authors should give a reference for methodology they used - hot plate and rotarod test.
3. Results:
- I recommend to repeat the rotarod test using higher speed. At 4 rpm it is unlikely to observe motor deficits induced by any drug tested. 10-24 rpm is more appropriate.
Line 138 and fig. 3 - p<0.05 or p>0.05?
- CIPN is manifested not only by thermal hyperalgesia but also tactile allodynia. Why was this effect not investigated? It is difficult to conclude that NO-711 is a promising agent for neuropathic pain treatment (lines200-201) if we do not know its impact on other than thermal hypersensitivity symptoms of neuropathic pain.
4. Discussion - some additional data regarding the effects of GAT1 inhibitors in other neuropathic pain models should be cited.
5. I suggest re-organisation of the metods and results - starting from pain tests and then rotarod test as an additional test.

Experimental design

All comments are mentioned in the section above.

Validity of the findings

According to my suggestion mentioned in Basic Reporting, point 3, which regards tactile allodynia, the discussion and the conclusions should be modified by clear statement of this study's limitation or the authors are requested to do some additional experiment.

Additional comments

No comments.

---

## Round 0.2 · accepted · Accept

Dear Authors,Congratulations your revised manuscript has now been accepted and will move into production.

·

Basic reporting

The submission meets the PeerJ standards

Experimental design

Experimental design is fine.

Validity of the findings

Data seems to be robust

Additional comments

The authors addressed all of my previous concerns